# TieBot: Learning to Knot a Tie from Visual Demonstration through a Real-to-Sim-to-Real Approach

**Weikun Peng[1], Jun Lv[2], Yuwei Zeng[1], Haonan Chen[3], Siheng Zhao[3],**
**Jichen Sun[2], Cewu Lu[2], Lin Shao[1†]**
[1] School of Computing, National University of Singapore
[2]Department of Computer Science, Shanghai Jiao Tong University
[3]Department of Computer Science and Technology, Nanjing University

**Abstract:** The tie-knotting task is highly challenging due to the tie's high deformation and long-horizon manipulation actions. This work presents *TieBot*, a Real-to-Sim-to-Real learning from visual demonstration system for the robots to learn to knot a tie. We introduce the Hierarchical Feature Matching approach to estimate a sequence of tie's meshes from the demonstration video. With these estimated meshes used as subgoals, we first learn a teacher policy using privileged information. Then, we learn a student policy with point cloud observation by imitating teacher policy. Lastly, our pipeline applies learned policy to real-world execution. We demonstrate the effectiveness of *TieBot* in simulation and the real world. In the real-world experiment, a dual-arm robot successfully knots a tie, achieving 50% success rate among 10 trials. Videos can be found on our [website](website).

**Keywords:** Learning from Visual Demonstration, Real-to-Sim-to-Real, Cloth Manipulation

## 1 Introduction

Learning cloth manipulation holds great utility across a wide range of applications. One intriguing domain is robotic tie knotting. Service robots must be adept at tasks like aiding the elderly or individuals with disabilities in dressing for certain social events. Teaching robots to knot ties, as a special case of cloth manipulation, typically pushes the limits of robotic cloth manipulation. This offers valuable insights for tie knotting and the broader field of robotic cloth manipulation.

Cloth manipulation presents challenges for robots due to its high-dimensional state and complex dynamics. Extracting and modeling state information are difficult problems. In contrast, humans have accumulated extensive knowledge about cloth manipulation. These priors make learning from demonstration (LfD) a promising direction. LfD empowers a robot to acquire a policy from expert demonstrations, significantly reducing the need to design task-specific reward functions manually. Consequently, LfD stands as a potent and efficient framework for instructing robots in the execution of complex skills.

However, existing LfD methods struggle with tie-knotting tasks. Kinesthetic demonstration or teleoperation suffers from the complexity of tie-knotting tasks. Tie-knotting tasks require bi-manual operations, placing high demand on human operators' skills and equipment. For instance, Zhang et. al use VR headsets for teleoperation [1]. Thus, simple behavior cloning may be significantly labor-intensive. Learning from visual demonstration is usually an easier approach in terms of collecting

---

† Correspondence to peng.weikun@u.nus.edu and linshao@nus.edu.sg

8th Conference on Robot Learning (CoRL 2024), Munich, Germany.

demonstration data. But this approach also leads to embodiment gaps. Therefore, researchers attempt to find some object-centric representations that robots can utilize to generate correct actions, overcoming embodiment gaps. Several methods attempt to learn a general visual representation of some simple pick-place skills via large-scale pre-training on actionless videos [2, 3, 4]. These works present strong generalizations on the learned visual representations, but none of them shows the ability to learn dexterous manipulation skills that can knot a tie. Other methods such as [5, 6, 7, 8, 9] try to leverage object trajectories or keypoints as representations to guide the policy learning. Such representations are indeed sufficient to describe simple object motions but fail to capture the tie's complex topology and subtle dynamics.

Compared to the existing LfD work mentioned in the previous paragraph, our insight is that mesh is the most suitable representation for tie-knotting tasks and other complex cloth manipulation tasks. It captures accurate geometric structures and physics properties of the tie, which is crucial for tie-knotting tasks. It also disentangles irrelevant information in the visual demonstrations, such as environment background, object colors, and so forth, enabling the learned policy to apply to different test settings. Therefore, inspired by [10], we propose a Real-to-Sim-to-Real LfD framework. First, we propose a Hierarchical Feature Matching method to iteratively estimate the tie's meshes with cloth simulation from the demonstrated video. We use a cloth simulator called *DiffClothAI* [11] that supports intersection-free contact for cloth to maintain the tie's topological structure during the estimation process. These estimated meshes from the demonstrated video are then used as subgoals. To learn where to grasp the tie and where to pull the tie from point clouds observations in simulation, we adopt a teacher-student training paradigm similar to [12]. Lastly, our pipeline executes learned policy in real-world settings.

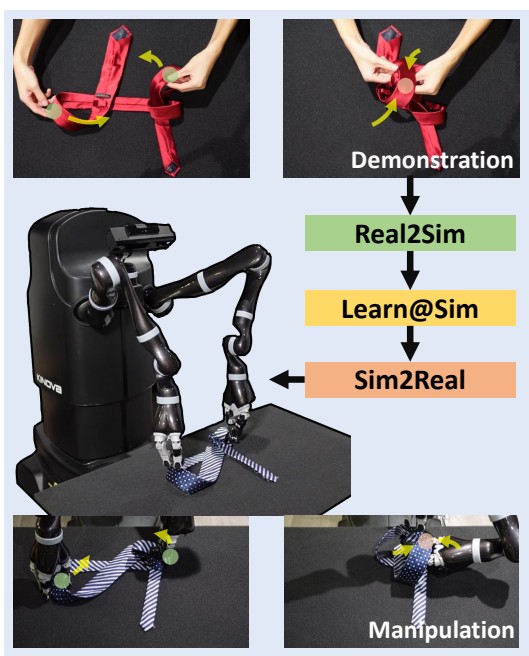

Figure 1: Our proposed *TieBot* performs a tie-knotting task. We leverage cloth simulation to recover the cloth's state from human demonstration and learn a goal-condition policy to accomplish the tie-knotting task.

In summary, we make the following contributions: 1) We introduced a systematic LfD framework for a dual-arm robot to learn to knot to tie. 2) We proposed a Hierarchical Feature Matching approach to estimate the tie's mesh with high deformation from the demonstrated RGB-D video using cloth simulation. 3) With estimated meshes as subgoals, we presented a teacher-student training paradigm to learn grasping points and placing points from point cloud observations in simulation. 4) We conduct experiments in simulation and the real-world to demonstrate the effectiveness and advances of our pipeline. To the best of our knowledge, this work is the first effort to develop a robotic system that integrates perception, modeling, and robot learning to train robots to tackle the task of tie-knotting—a particularly complex and underexplored area of cloth manipulation.

## 2 Related Work

### 2.1 Cloth Manipulation

Previous work mainly addresses short-horizon cloth manipulation tasks that only involve simple pick-place actions. There are several approaches to learning cloth manipulation skills. One approach

is using model-free RL or learned dynamics model to learn cloth unfolding, rope rearranging, and dressing assistance tasks on raw sensor input [13, 14, 15, 16, 17]. Other approaches will collect and annotate data from images [18, 19] or generate demonstration trajectories in simulation [20] to learn policy. Because of the short-horizon and simple actions features of tasks, it's also possible to infer correct actions from some visual representations, such as flow between current observation and target images [21].

In contrast, tie-knotting tasks require flipping or rotating a part of the tie, which makes it difficult to annotate robot actions or design action primitives. Therefore, collecting and annotating robot actions on observations is infeasible. It's also difficult to generate demonstrations or directly apply RL in simulation since the trajectories of tie-knotting tasks are much longer and the possible state space is much larger. Thus, in this work, we choose to learn skills from human demonstration.

## 2.2 Learning from Visual Demonstration

One line of research explores pre-training neural representations from actionless videos [22, 23, 2, 24, 3, 4, 25]. This approach aims at learning general representations for different actions, whereas none of them shows the ability to learn dexterous manipulation skills that can knot a tie. Another line of research attempts to learn from visual priors extracted from visual demonstrations, such as object trajectories [5, 26], hand poses [27, 28], keypoints positions [29, 6], graph relations [7], or affordances [30]. The third approach is to learn a video or trajectory prediction model to guide policy learning [31, 32, 33, 34, 8]. These approaches require in-domain demonstrations, placing restrictions on visual demonstrations. Moreover, the prediction model may suffer from the long-horizon feature of tie-knotting tasks. ORION is the most closely related work, which builds a graph representation from object motions that can generalize across diverse test environments [35]. However, simple graph representations cannot capture the tie's complex topology and subtle dynamics during the tie-knotting process.

Consequently, we propose explicitly modeling the demonstration as a sequence of meshes. Mesh can accurately describe the tie's structure and dynamics, which is crucial to learning correct robot actions and generalizing them to different test scenarios.

## 2.3 Cloth State Estimation

One cloth state estimation method directly predicts cloth states using deep learning [36, 37, 38, 39]. Non-rigid point cloud registration methods such as coherent point drifting are also applied for linear deformable object tracking [40, 41, 42]. However, purely vision-based methods do not guarantee correct cloth topology due to the lack of physics prior. Huang et al. propose a method to reconstruct and track cloth state with a dynamics model [43]. However, this method requires known actions, which cannot be accessed easily from human demonstration sometimes. Lv et al. use differentiable rendering to estimate the state of linear deformable objects [10]. However, tie is a 2D deformable object.

Therefore, we propose a Hierarchical Feature Matching method to iteratively estimate the tie mesh in the demonstration video with cloth simulation. Cloth simulation provides important physics prior for state estimation, such as non-penetration, which is crucial for maintaining correct topology.

## 3 Technical Approach

This work presents a Real-to-Sim-to-Real LfD framework called *TieBot* to guide a dual-arm robot shown in Fig. 1 to knot the tie from an RGB-D demonstration video. An overview of our proposed method is in Fig. 2. We first describe the procedure to estimate the tie's mesh sequences from the demonstrated video (Sec. 3.1). Using the tie's mesh sequences as subgoals, we introduce a pipeline to generate robot actions to manipulate the tie, using teacher-student training paradigm (Sec. 3.2).

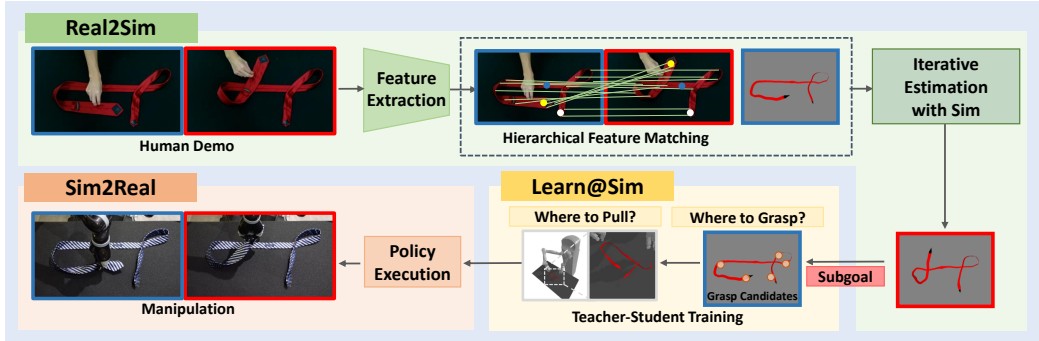

Figure 2: *TieBot* utilizes simulation to estimate the tie's meshes from the demonstrated video. Then, using mesh sequences as subgoals, we introduce how to generate the robot's actions to manipulate the tie. The pipeline finally executes learned policy in real world.

## 3.1 Real2Sim

To better estimate the tie's mesh, we propose to integrate cloth simulation into our pipeline, which provides important physical prior such as non-penetration in the estimation process. We segment the tie in the demonstrated RGB-D video using *Track-Anything* [44] and transform the associated segmented depth images into point clouds. Meanwhile, a tie's mesh is loaded into the *DiffClothAI*. At time step $t$, we use the tie mesh's vertices denoted as $\mathcal{Y}_t^S$ to describe the tie's shape. From the RGB images and segmented point clouds denoted as $\{\mathcal{I}_t^D\}$ and $\{\mathcal{X}_t^D\}$, Real2Sim pipeline estimates tie's mesh sequences $\{\mathcal{Y}_t^S\}$ with simulation. The pipeline manually aligns the mesh with the initial frame. We assume the initial mesh fully overlaps with the initial point cloud.

**Local Feature Matching.** If $\mathcal{Y}_{t-1}^S$ and $\mathcal{X}_{t-1}^D$ are aligned and there are correspondences between $\mathcal{X}_{t-1}^D$ and $\mathcal{X}_t^D$, we can build the correspondences from the tie's mesh to the next demonstrated point cloud $\mathcal{X}_t^D$ and move the tie's vertices to align $\mathcal{Y}_t^S$ towards $\mathcal{X}_t^D$.

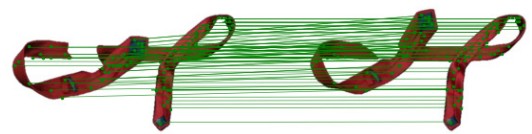

Here we adopt an off-the-shelf feature matching model called *LoFTR* [45] to build up correspondences between two RGB images $\mathcal{I}_{t-1}$ and $\mathcal{I}_t$ as shown in Fig 3. Typically, *LoFTR* can

Figure 3: Local Feature matching between two images. A hand caused a gap along the length of the tie during the demonstration.

provide more than a hundred reliable correspondences between two images, which cover almost every visible part of the tie. From the correspondences between $\mathcal{I}_{t-1}$ and $\mathcal{I}_t$, we can find the feature points on $\mathcal{X}_{t-1}^D$ and their corresponding feature points on $\mathcal{X}_t^D$. Then, to control the mesh in *DiffClothAI* to align it with $\mathcal{X}_t^D$, we need to define several vertices on the mesh as control vertices $\mathcal{V}$. Since $\mathcal{Y}_{t-1}^S$ aligns well with $\mathcal{X}_{t-1}^D$, we map the feature points on $\mathcal{X}_{t-1}^D$ to nearest vertices on $\mathcal{Y}_{t-1}^S$. These vertices are assigned as control vertices $\mathcal{V}$. Finally, we control $\mathcal{V}$ to move to the positions of feature points on $\mathcal{X}_t^D$ to align $\mathcal{Y}_t^S$ towards $\mathcal{X}_t^D$ in *DiffClothAI*.

However, vanilla local feature matching cannot create correspondences in occluded regions, which is common in tie-knotting tasks. We lose the motion information due to occlusion, and the estimation will deviate. Therefore, we propose to add global keypoints information to amend this pipeline.

**Keypoints Detection.** Keypoints detection can directly build correspondences between mesh vertices and the point cloud. Thus, it will not be affected by occlusion. We define five keypoints along the tie's surface and the corresponding five key vertices on the mesh, shown in Fig. 4. For each keypoint as the origin, we define the local frame as follows. The z direction is the surface normal from the tie's positive side to the negative side. The x direction is the direction of the tie's middle skeleton. The y direction is derived using the right-hand rule. These five keypoints, in a predefined order, play the role of the skeleton definition.

Then, we train Pointnet++ [46] to predict the keypoints and associated local frames on the demonstrated point clouds. However, the high-dimensional state makes it challenging to generate sufficient training data to cover all the states encountered in the knotting procedure. A successful tie-knotting trajectory occupies only

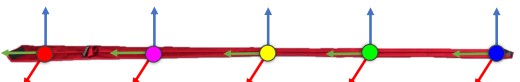

Figure 4: The oriented keypoints to represent the state of the tie. The x,y,z axis are represented by the red, green, blue arrow, respectively.

a small portion of the whole state space of the tie. Thus, uniformly applying random actions on the initial tie's mesh in the simulation to produce training data fails to cover these states. In contrast, we generate training data based on the current estimated mesh. When we detect the chamfer distance between $\mathcal{Y}_t^S$ and $\mathcal{X}_t^D$ is larger than a threshold, we backtrack to the previous time step, gather the tie's shape $\mathcal{Y}_{t-1}^S$ and apply random actions to the tie's mesh at $t-1$ in the simulation to generate annotated training data and train the keypoints prediction network.

**Hierarchical Feature Matching (HFM).** Finally, we combine them as Hierarchical Feature Matching (HFM) for state estimation. Control vertices assigned in local feature matching and key vertices will be used together to pull the mesh to target positions specified by local feature matching and keypoints detection. Local feature matching provides detailed motion of vertices, while global keypoints indicate a global tie's structure. This global structural information ensures the estimation won't deviate too much, alleviating the shortcomings of the local feature matching method. We use this method to estimate the tie's meshes from demonstration and output a sequence of meshes $\{\mathcal{Y}_t^{\mathcal{G}}\}$. The next part will use these meshes as subgoals to guide robot action generation.

## 3.2 Learn@Sim

Our pipeline begins to sequentially generate feasible robotic actions in the simulation to guide the tie $\{\mathcal{Y}_t^S\}$ towards these subgoals. Since the tie-knotting task is a long-horizon task with multiple grasp and pull actions, we aim to learn where to grasp the tie and where to pull the tie.

For where to pull, we apply a simple strategy. Once we identify the grasping vertices, we pull these vertices to the positions of those vertices on the subgoal. For where to grasp, we adopt a similar teacher-student training paradigm in [12] to ease policy learning. Directly learning from high dimensional observations such as point cloud is data-inefficient because the policy needs to simultaneously learn which features to extract from visual observations and what the high-rewarding actions are. On the contrary, learning a policy via RL from sufficient state information would be much easier, as suggested by [12]. Therefore, we first use privileged information to learn a teacher policy, and then train a student policy imitating teacher policy with point clouds as observations.

**Teacher Policy.** We first learn a teacher policy to select proper grasping points using privileged information. The state $s$ contains the previous tie's vertices positions and the point-wise displacement for each tie vertices to the subgoal. The action $a$ is one or two grasping vertices of all the tie mesh's vertices. The reward function $\mathcal{R}$ is defined in equation 1.

Note that we specify the action space as the discrete space (vertex index of the tie). Although there are multiple 6D poses of the robotic grippers to grasp one vertex position of the mesh, the learned policy still reflects the overall grasping quality of these 6D poses associated with one vertex. In the engineering practice, we record each grasping pose offline so that once we figure out the grasping vertices on the tie's mesh at each timestep, we can automatically produce the feasible grasping poses concerning specific hardware platforms using inverse kinematics.

$$\mathcal{R}(s, a) = \begin{cases} C_1, & \text{if knotting-tie succeeds} \\ -C_2, & \text{if fails to reach any subgoal along the trajectory} \\ C_3 - \|\mathcal{Y}_t^S - \mathcal{Y}_t^{\mathcal{G}}\|, & \text{Otherwise} \end{cases} \quad (1)$$

For the reward function, here $C_1, C_2, C_3$ are constant positive values. $\mathcal{X}_t^S$ is the result tie mesh. The failure to reach the subgoal is due to the distance $\|\mathcal{Y}_t^S - \mathcal{Y}_t^{\mathcal{G}}\|$ is larger than a given threshold, the tie

could not be pulled close to the subgoal by grasping on the wrong selected vertex; otherwise, it will return $C_3 - \|\mathcal{Y}_t^S - \mathcal{Y}_t^{\mathcal{G}}\|$ for intermediate steps or $C_1$ for the final step.

**Student Policy.** To learn actions from point clouds, we train a student policy to imitate teacher policy. We add some perturbations to the size and positions of the mesh and update the associated trajectories accordingly to generate training data in the simulation. We render point clouds from meshes in PyBullet [47] as the input of our policy network $\pi^{sim}$ and output the grasping points and placing points positions. We use Pointnet++ [46] as the policy network and train it in a supervised learning manner.

## 4 Experiments

In this section, we introduce our experimental setup and conduct quantitative and qualitative evaluations to demonstrate the effectiveness of our approach. Our experiments focus on answering the following questions.

- How do our pipeline and baseline methods perform on tie-knotting task?
- Can our pipeline apply to other cloth manipulation tasks?
- How does HFM compare to other cloth state estimation methods?
- Can our HFM apply to other cloth manipulation tasks?

Considering the complexity of the entire system, we provide additional experiment results, along with detailed explanations of submodules, in the supplementary materials and website.

### 4.1 Comparing *TieBot* and Baseline

We first evaluate the whole pipeline of *TieBot* and a baseline method in a tie-knotting task. We estimate a sequence of meshes from one human demonstration video. Then, we divide the whole trajectory into 6 parts with 6 subgoals. Our teacher policy learns to select proper grasping points using PPO [48], and student policy imitates the teacher policy to infer grasping points and placing points from the point cloud. We evaluate *TieBot* and the baseline method 10 times for each of the two different ties in *DiffClothAI* and evaluate *TieBot* on two real ties with a dual-arm robot.

To illustrate our pipeline applies to other cloth manipulation tasks besides the tie-knotting task, we conduct experiments on the towel-folding task in the real setting. The towel-folding process is shown in the last row of Fig. 7.

**ATM.** ATM proposes to model tasks as points trajectories [8]. It first learns a trajectory prediction model, and then learns policy with the learned prediction model using imitation learning. For tie-knotting task, following similar experiment settings in ATM, we collect 100 demonstration videos in simulation to train the trajectory prediction module. Then, we use the 45 demonstration videos

| Success Rate / Average Achieved Subgoals | Ours | ATM |
|---|---|---|
| simulation: normal tie | **60% / 5.1** | 0% / 0.0 |
| simulation: larger tie(unseen) | **30% / 4.3** | 0% / 0.0 |
| real: real tie(softer) | 50% / 5.0 | NA / NA |
| real: real tie(harder, unseen) | 30% / 4.15 | NA / NA |
| real: towel(unseen) | **70% / 1.6** | 0% / 0.8 |

Table 1: Success rate and average achieved subgoals of policy rollouts

with ground truth action annotations to train the policy network and test the policy in simulation. The action is the 3D offset of the grasping vertices. For towel-folding task, we collect 100 human demonstrations and 10 robot demonstrations in the real world. We train the ATM to predict the displacement of the two robot arms' end effectors in cartesian space.

**Metrics.** We compare the success rate between our pipeline and ATM [8]. For the tie-knotting task, in simulation experiments, if the distance of the final tie's mesh to the target tie's mesh is smaller than a threshold, we consider it a success. In real-world experiments, if the little end of the tie is inserted

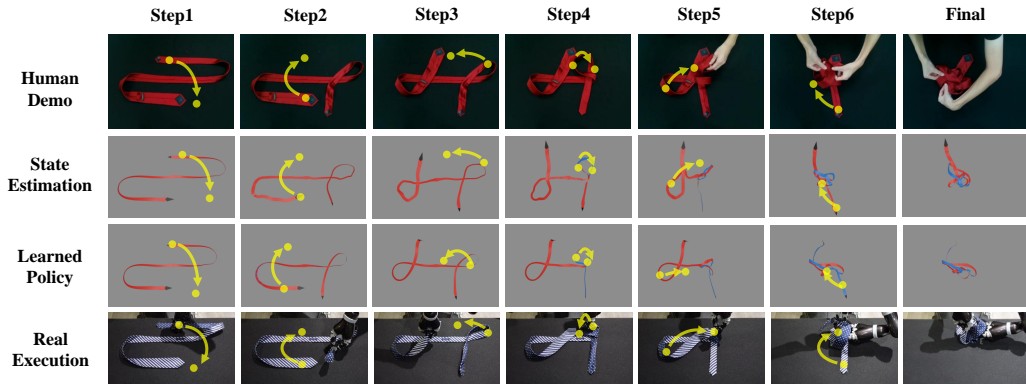

Figure 5: The results of *TieBot* at different stages. We show different sides of the tie in red and blue and manipulation action in yellow to better visualize.

into the hole, as shown in the final stage in Fig. 5 and Fig. C.9, we consider it a success. Since the tie-knotting task is long-horizon, we also compute the averaged number of achieved subgoals for further evaluation. For the towel-folding task, We consider the state to be successful if the towel stays totally on one side of the folding line, as shown in Fig. D.12.

**Experiment Result.** We test *TieBot* on two real ties that differ in materials: one is softer, and the other one is harder. We tested each of them 10 times. We also test *TieBot* and ATM on two ties with different sizes in simulation 10 times for each. The quantitative results are shown in Tab. 1, and qualitative results of *TieBot* are shown in Fig. 5. This comparison suggests that object trajectories are insufficient to represent subtle dynamics and topology of the tie in tie-knotting tasks. ATM quickly deviates from the correct trajectory since it cannot capture the subtle dynamics of the tie. Therefore, it fails to achieve even one subgoal. Hence, explicitly modeling the tie in meshes is necessary. For qualitative evaluation, in Fig. 5, we can see that although the tie in the demonstration video, the mesh in the simulation, and the tie used for real robot manipulation are different, our pipeline can overcome these gaps and learn feasible robot policy. For the towel-folding task, we find that ATM can learn the first folding action but struggle with learning the latter action.

## 4.2 Evaluating Hierarchical Feature Matching (HFM)

Real2Sim is the most important part of our pipeline. Without accurate state estimation, particularly estimating the correct topology for the tie, it's impossible to learn a feasible policy to accomplish the task. To illustrate the importance of different components of HFM and its performance against other cloth state estimation methods, we design three experiments in simulation to test baseline methods and the ablation versions of HFM.

**Coherent Point Drift.** Coherent Point Drift (CPD) [49] is a non-rigid point cloud registration algorithm. We employ the CPD to predict the target positions of the mesh vertices in the target point cloud and directly align the mesh to the target positions.

**Ablated Version.** *Ours w/o KP* stands for only using local feature matching; *Ours w/o LF* stands for using local feature matching and the predicted keypoints positions; *Ours w/o FM* stands for only using predicted keypoints positions and local frames.

| L2 Distance | Ours | Ours w/o FM | Ours w/o KP | Ours w/o LF | CPD |
|---|---|---|---|---|---|
| exp1 | **0.0248** | 0.0732 | 0.2424 | 0.0512 | 0.1384 |
| exp2 | **0.0053** | 0.0107 | 0.0123 | 0.0088 | 0.0661 |
| exp3 | **0.0032** | 0.0093 | 0.0053 | 0.0049 | 0.1049 |

Table 2: Quantitative results of ablation study and comparison to CPD in simulation.

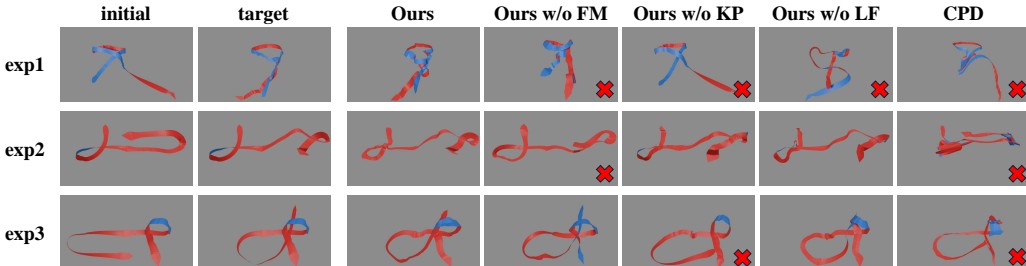

Figure 6: The visualization of the ablation study of HFM in simulation. We put a cross sign in the image's bottom-right corner to indicate failures of estimating the correct target state, according to human evaluation. Red and blue colors represent different sides of the mesh.

**Experiment Result.** The qualitative results are shown in Fig. 6. We can find that either CPD or ablated versions of HFM cannot estimate the target mesh correctly among these three experiments. We also compute the L2 Distance between the vertices of the target mesh and estimated mesh as a quantitative evaluation shown in Tab. 2. It also suggests that the performance will degrade if we cancel some parts of HFM, while CPD deviates a lot from the correct states.

### 4.3 Apply HFM on Other Cloth Manipulation Tasks

We demonstrate that HFM can be applied to other cloth manipulation tasks. One is a different way to knot a tie. The other one is to fold a towel. We visualize the estimation results in Fig. 7. The results show that HFM can be applied to different cloth manipulation tasks.

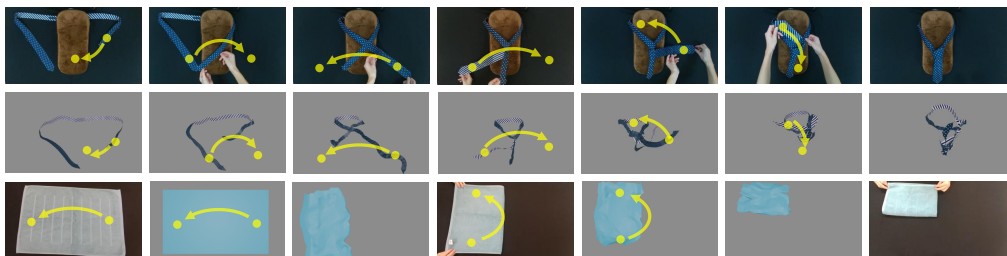

Figure 7: The visualization of a different way to knot a tie and the towel folding. The first row is the human demonstration of tie-knotting, the second row is the estimated states in simulation, and the third row is towel folding. We show the manipulation action in yellow dots and arrows.

## 5 Conclusion

This paper presents TieBot, a novel system designed to teach robots to perform the tie-knotting task through a Real-to-Sim-to-Real learning approach. The framework leverages visual demonstrations and integrates a mesh of the tie to capture its complex structure and subtle dynamics, which are essential for effective manipulation. Through the proposed Hierarchical Feature Matching method, we estimate tie meshes from demonstration videos and use them as subgoals for policy learning in simulation. Our teacher-student training paradigm enables robots to learn from point cloud observations and execute the learned policy in real-world settings. TieBot demonstrates promising results both in simulation and real-world experiments, achieving a 50% success rate for tie-knotting. Additionally, we showcase its potential for other cloth manipulation tasks, such as towel folding, indicating the broader applicability of our approach. This work marks a significant step forward in robotic cloth manipulation, particularly for long-horizon, complex tasks such as tie-knotting.

Nonetheless, our pipeline has some limitations. First, our Real2Sim module requires training key-points detection models iteratively, which is computationally intensive. Second, due to the hardware limits, the last step in the real-world experiments shown in Fig. 5 is hardcode action. Better video tracking methods and more dexterous robot arms may alleviate these issues.

**Acknowledgments**

The authors would like to thank Zihao Xu from National University of Singapore for setting up the tie-knotting experiment in the real setting, Zhixuan Xu and Haoyu Zhou from National University of Singapore for the support of computation resources, Hongjie Fang from Shanghai Jiao Tong University and Flexiv Robotics for helping with towel-folding experiment in the real setting.

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

# Appendix

## A    More Discussion on Real2Sim

### A.1    Why not Use Video Tracking?

One may be curious about why not use video tracking to extract a tie's motion in our work. We test co-tracker [50] and DINO-Tracker [51] on our tie-knotting demonstration videos. For instance, in the towel-folding task, we tested DINO-Tracker on the demonstration video. From the results shown in Fig. A.1, we can find that even the state-of-the-art video tracking model cannot provide accurate point-level trajectory.

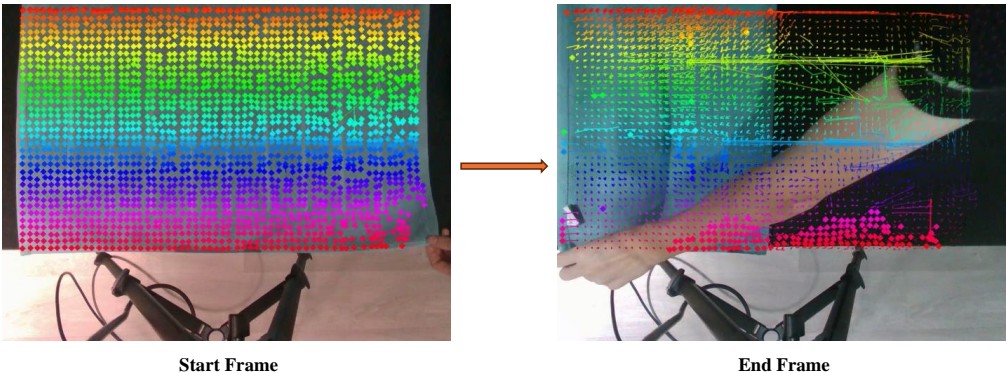



**Start Frame**          **End Frame**



Figure A.1: Track prediction results of the towel-folding task.

### A.2    Experiment on Out-of-Distribution Issue of Keypoints Detection

To illustrate the necessity of iteratively updating the detection model, we train a neural network with the same structure on a randomly sampled tie's shape based on the initial shape. We trained our iterative keypoint detection model on 14 different shapes separately. For each shape, we randomly generate 500 similar shapes for training. For the randomly sampled method, we randomly generate 7000 shapes from the initial tie's shape to train a single keypoint detection model. We compare the result of **Ours** with random sample **RS** result in Fig. A.2. Compared to the human annotation result, **RS**'s predictions have larger errors than **Ours**. **RS** method encounters an out-of-distribution problem. The test shape of the tie cannot be easily sampled, so **RS** cannot generalize to this test case.

We also test the iterative global keypoint detection and **RS** method in simulation with ground truth annotation. The quantitative results are listed in Tab. A.1.

|      | Position Error(m) | Z-axis Error(°) | X-axis Error(°) |
|------|-------------------|-----------------|-----------------|
| *Ours* | **0.028**       | **10.68**       | **14.41**       |
| *RS*   | 0.183           | 49.00           | 68.76           |

Table A.1: Quantitative results of iterative keypoint detection on simulation data

### A.3    Implementation Details of Keypoints Detection

#### A.3.1    Keypoint Positions Prediction

**1) data generation**    We first load the mesh model into *DiffClothAI* [11], choose keypoint as control vertices, and apply random perturbations to these vertices to generate different shapes of the tie. Then, we load generated mesh models into *PyBullet* [47] to render point clouds. We use the same

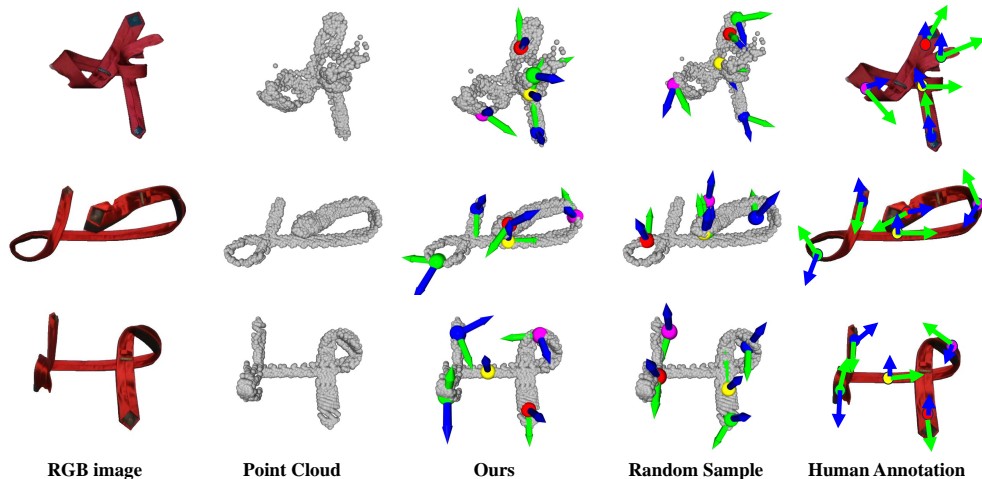

| RGB image | Point Cloud | Ours | Random Sample | Human Annotation |

Figure A.2: Prediction results of the oriented keypoints on real image and point cloud.

camera intrinsics as demonstrated video and similar camera pose to render point clouds. We generate 500 point clouds as training data.

To annotate training data, we first compute the geodesic distance of all points in the point cloud to keypoints. Then, we convert the geodesic distance to probability using equation 1. $d$ is the geodesic distance, $\sigma$ is a hyperparameter, $p$ is the probability to evaluate how likely the point is to be a keypoint. In our experiments, we use $\sigma = 0.15$. Thus, if a point is close to one of five keypoints, the probability corresponding to that keypoint will be high.

$$p = \exp^{-\frac{d^2}{2\sigma^2}} \tag{1}$$

**2) training details** Different from the original pointnet++ semantic segmentation model [46], we change the last layer to sigmoid. The training parameters are listed in Tab. A.2.

| parameter name | parameter value |
|---|---|
| loss function | L2(for keypoint positions and offsets prediction) |
| | L1(for normal and middle line direction prediction) |
| data augmentation | gaussian noise, random scale, random rotation |
| training epochs | 80 |
| batch size | 24 |
| learning rate | 1e-4 |
| optimizer | Adam |
| scheduler | cosine annealing with 10 epochs warm-up |

Table A.2: Hyperparameters for training global keypoint prediction

**3) inference details** Our model takes a point cloud as input and outputs a probability matrix $P \in (0,1)^{N \times 5}$, $N$ is the number of points in the point cloud. Each entry $P_{i,j}$ represents the probability of point $i$ to be keypoint $j$. To decode the predicted keypoints positions, we first select points with the top 5% probability as inlier for each column of $P$. Then, we assign other points' probability to zero and normalize the probability for each column of $P$. Now we get the normalized probability distribution of each keypoint, denoted as $\hat{P}$. Finally, we compute the average positions of all points weighted by normalized probability, $x_k = \sum_{i=1}^{N} \hat{P}_{i,k} \cdot x_i$. This is the final prediction for keypoint positions.

### A.3.2   Normal(Z Axis) Prediction

**1) data generation**   We generate data the same way as keypoint positions. For annotation, we first compute the normal direction of each face of the mesh. Then, we assign these values to points in the point cloud according to the nearest faces.

**2) training details**   We also remove the log_softmax layer in pointnet++ [46]. We use L1 distance as the loss function. The other training parameters are the same as keypoint position prediction.

**3) inference details**   With predicted keypoints positions and predicted normal directions of all points, we compute the normal of each keypoint as the average of neighboring points normal directions.

### A.3.3   Middle Line(X Axis) Prediction

**1) data generation**   Same as normal prediction, just change the annotation from normal direction to middle line direction.

**2) training details**   It's the same as normal prediction.

**3) inference details**   It's the same as normal prediction.

### A.4   Results of Local Feature Matching and Keypoints Detection on Real Data

We present some examples of local feature matching and keypoints detection on two tie-knotting tasks and a towel-folding task, shown in Fig A.3.

### A.5   Ablation Study of HFM on Real Data

We demonstrate the effectiveness of hierarchical matching in cloth state estimation in Fig. A.4. In the first test case, we aim to illustrate the importance of global keypoint detection. Therefore, we chose two images that show differences in positions and orientations. In the third column, **Ours** method successfully flips the tie and moves forward a little, as shown in the images. **Our w/o KP** and **Ours w/o LF** cannot move forward as expected. Because, in this case, local feature matching cannot find correspondences in the tie's left part.

In the second test case, we aim to illustrate the importance of local feature matching. We chose two images that contain an operation of lifting a side of a ring in the air. This action requires detailed information to achieve accurate estimation. The last column shows the result of **Ours w/o FM**. Our framework cannot accurately estimate the shape only with global keypoint positions and local frames. This global information can only provide general structure guidance instead of detailed shape information.

## B   More Discussions on Learn@Sim

### B.1   How to Control Tie in *DiffClothAI*

Modeling grasping in *DiffClothAI* [11] is not simply selecting one vertex on the mesh as the control vertex. Because controlling one vertex is not enough to simulate rotation in *DiffClothAI*, knotting a tie requires some rotation actions. Therefore, we select one central vertex and its surrounding vertices as control vertices, shown in Fig.B.5. By controlling a small region instead of a single vertex, we can simulate rotation actions in *DiffClothAI*.

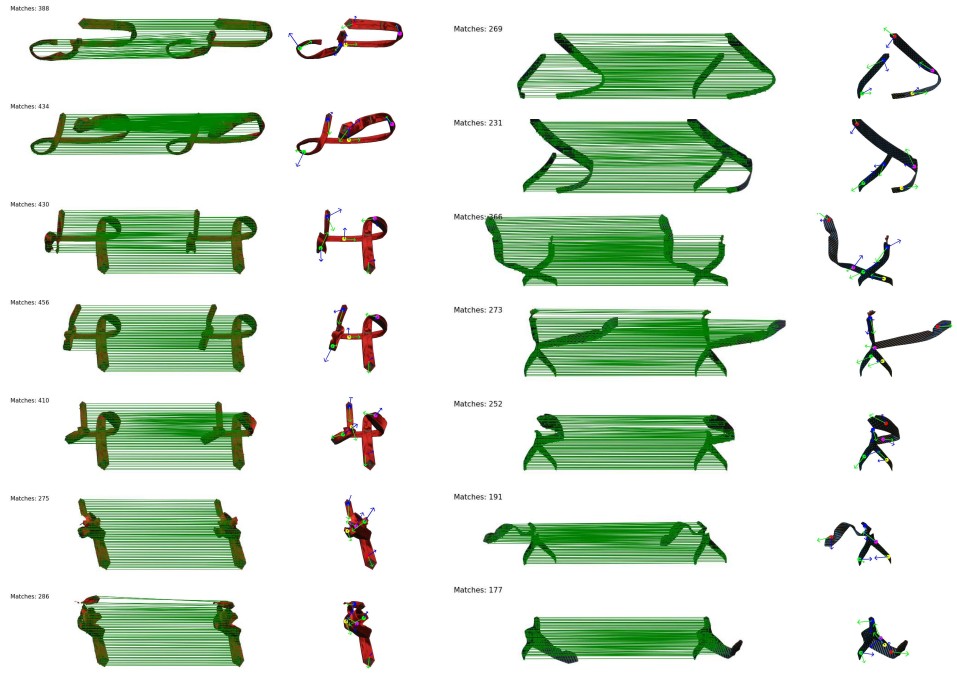

(a) Local feature matching and keypoints detection results on real-world tie-knotting demonstration

(b) Local feature matching and keypoints detection results on another real-world tie-knotting demonstration

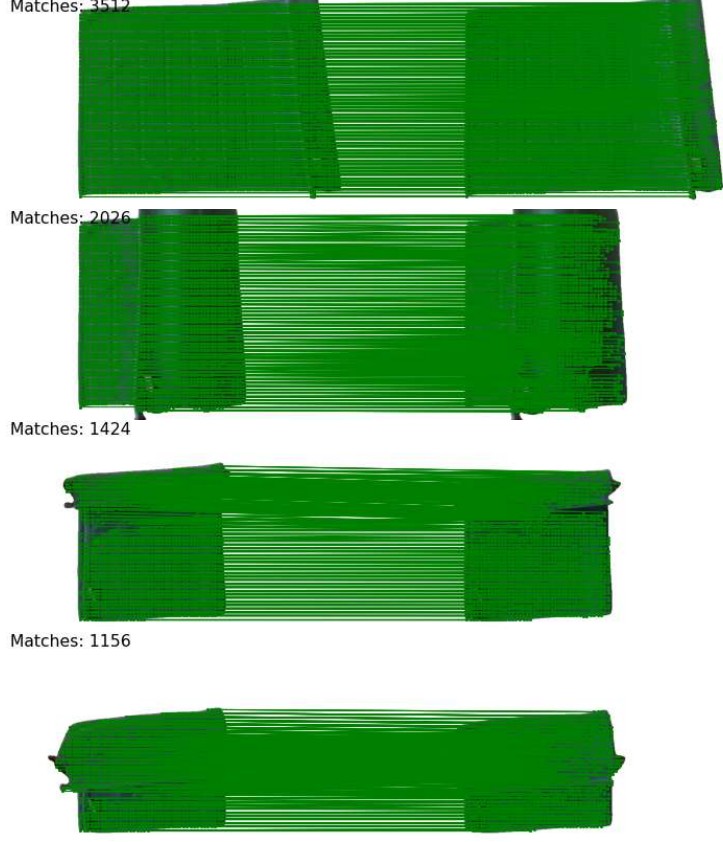

(c) Local feature matching results on another real-world towel-folding demonstration

Figure A.3: We test local feature matching and keypoints detection on real-world demonstrations. It shows that our method works for most tie shapes.

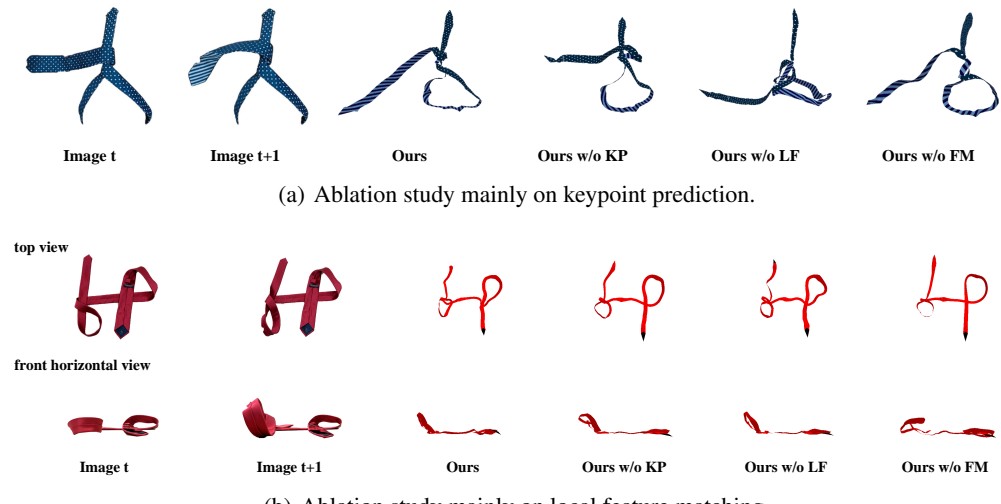

(a) Ablation study mainly on keypoint prediction.

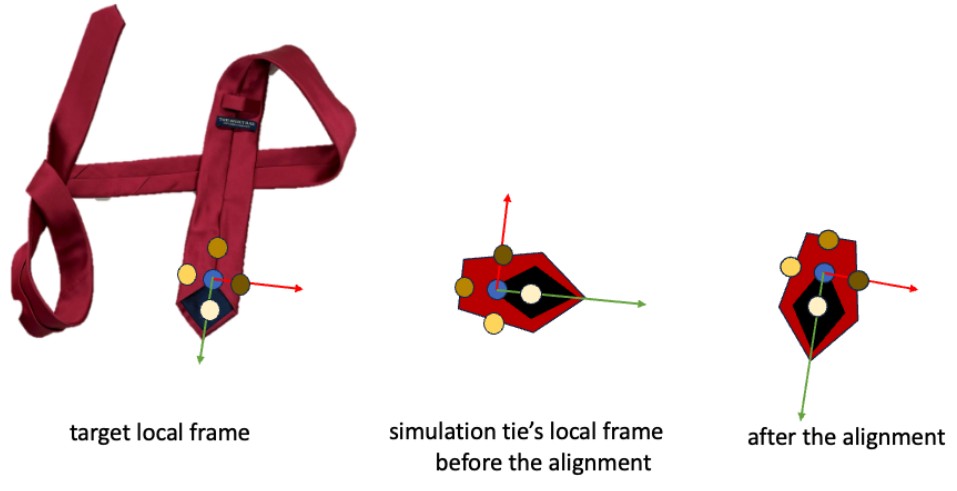

(b) Ablation study mainly on local feature matching.

Figure A.4: Ablation study on hierarchical feature matching for state estimation.

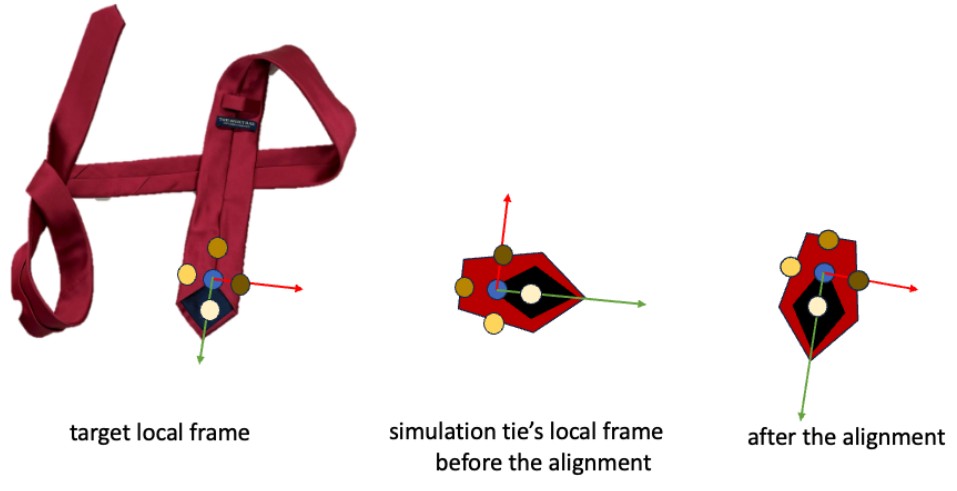

Figure B.5: Illustration of control vertices in *DiffClothAI*.

## B.2 How are the grasp and pull actions defined

**Grasp Actions** For the grasp action, we begin by sampling a set of vertices from the mesh, which serve as grasping vertex candidates. The grasp action is then defined as selecting different combinations of these vertices. This selection process is treated as a discrete action space, where the policy chooses the optimal vertices for grasping based on the current state of the tie.

**Pull Actions** The pull action is defined as the target position for the robot's gripper. Once the policy has selected the grasping vertices at a particular step, the corresponding goal positions are determined by the positions of these vertices on the subgoal mesh. This approach means that the pull action is not learned directly; instead, it is derived automatically based on the selected grasping points and the configuration of the subgoal mesh, leveraging the differentiable cloth simulation.

### B.3 Implementation Details of Teach-Student Training Paradigm

#### B.3.1 Teacher Policy

We model the grasping point selection as MDP and use model-free RL to learn the proper grasping point. To simplify the problem, we sample 40 vertices on the middle line of the mesh model as our candidates. vertices directly connected to these candidates in left, right, up, and down are defined as their neighbors.

In practice, we evenly sample 40 vertices on the middle line of the mesh model as our candidates. The state $s$ is a $40 \times 6$ matrix. The action $a$ is a $820 \times 1$ one-hot vector, which contains grasping one vertex(40) and two vertices($40 \times 39/2 = 780$).

To learn to select grasping points, we use PPO implemented in stable-baseline3 [52]. The hyperparameters are shown in Tab B.3 for all trajectories.

| parameter name | parameter value |
|---|---|
| learning rate | 0.0003 |
| batch size | 64 |
| $\gamma$ | 0.99 |
| gae_lambda | 0.95 |
| clip range | 0.2 |
| $C_1$ | 5 |
| fitting threshold | {0.9, 1.5, 2.0, 3.0, 3.0, 1.9} (listed in subgoals order) |
| $C_2$ | 30 |
| $C_3$ | 30 |

Table B.3: Hyperparameters for learning grasping points settings

#### B.3.2 Student Policy

To learn student policy, we first execute the teacher policy multiple times within the DiffClothAI simulation environment. During each run, the teacher policy uses the current mesh and the subgoal mesh to predict the selected grasping vertices on the current mesh, as well as the corresponding placing points. Through these simulations, we generate a dataset comprising 3,000 mesh-grasping-placing pairs, which serve as the foundation for training the student policy.

To simulate real-world conditions, we load all the generated meshes into PyBullet and use the same camera intrinsics as those used in our real-world setup to render point clouds. This ensures that the point clouds accurately reflect the observations a robot would receive in practice.

The student policy is then trained using supervised learning, where the input is the rendered point cloud and the output is the predicted grasping points and placing points. The grasping and placing points produced by the teacher policy serve as the ground truth labels for this training process. The student policy learns to map point cloud observations to the appropriate actions (grasping and placing points), effectively mimicking the decisions made by the teacher policy based on the point cloud data alone.

The training details are the same as training keypoints prediction, only changing the number of keypoints from 5 to 2. We follow the same training hyperparameters as grasping point prediction for placing point position detection, only changing the pointnet++ semantic segmentation model to the classification model.

#### More Results on ATM Baseline

We first illustrate example outputs of ATM baseline on two different ties in Fig. B.6. We can see that without explicit mesh modeling, ATM will quickly deviate from correct trajectories.

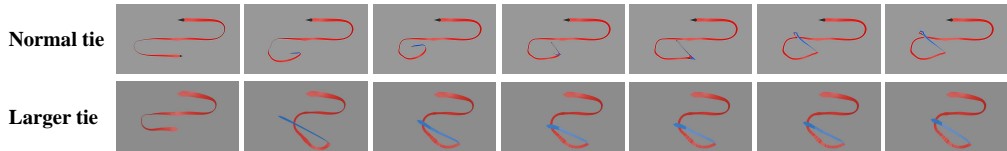

Figure B.6: Illustration of ATM rollouts in simulation.

To further examine whether the long-horizon property or points trajectories representations lead to the failure of ATM on tie-knotting tasks, we further conduct experiments on some shorter tasks to see if ATM can work. Specifically, we divide the whole tie-knotting task into 6 subtasks, training and testing ATM on each subtask separately. The results are shown in Fig. B.7. We can see that ATM's results look better for the first 3 subtasks. But ATM still cannot complete the last 3 subtasks, which involve complex topology and subtle dynamics. This experiment demonstrates that using points trajectory representation cannot handle such complex tasks even with a shorter horizon.

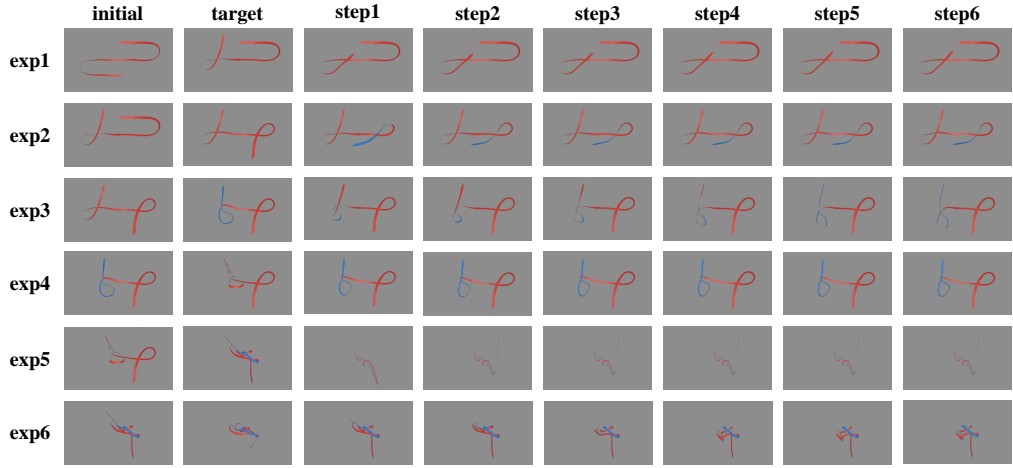

Figure B.7: Illustration of ATM results on 6 subtasks.

## C   More Discussions on Real-World Experiment

### C.1   Real-World Experiment setup

We set up the real-world experiment with a dual-arm robot as shown in Fig. C.8. The MOVO robot [53] has two 7 DoF arms and a Kinect RGB-D camera overhead. We perform position controls and use RangedIK [54] for solving inverse kinematics. The success state is defined in Fig. C.9.

### C.2   Failure Cases and Analysis

Two major failure cases in real-world experiments are shown in Fig. C.10. One is the robot fails to rotate the whole ring structure of the tie, another is the robot fails to insert the little end of the tie into the whole shown in Fig. C.9.

The first case is caused by the subtle dynamics of the tie. To rotate the ring structure, the tie should be a bit harder so that the ring structure will not crumple during the rotation process, while it should not be so hard so that the rotation action won't interfere with other parts of the tie. This places a high demand on both the tie and the robot. It's hard to solve from the algorithm side.

The second case is caused by partial observation of this task. We use one camera on the top of the robot for perception. It cannot perceive the little end of the tie in this case. Thus, the robot has to act blindly, lowering the success rate.

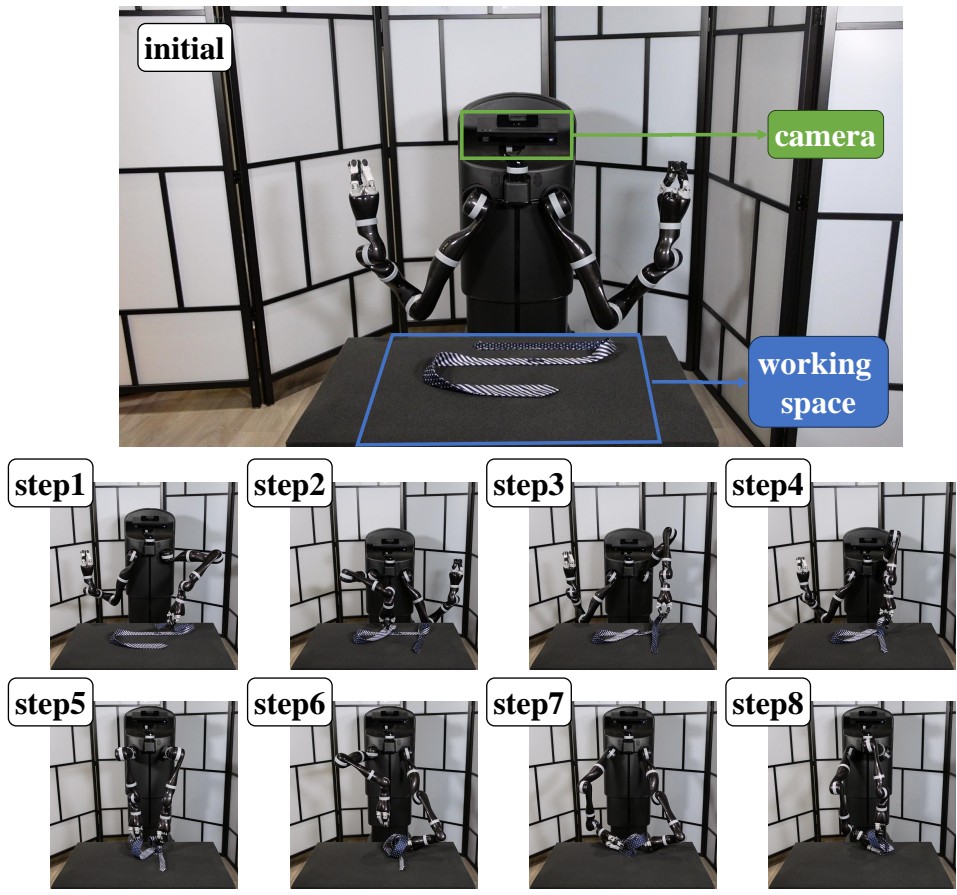

Figure C.8: Illustration of real-world experiment settings.

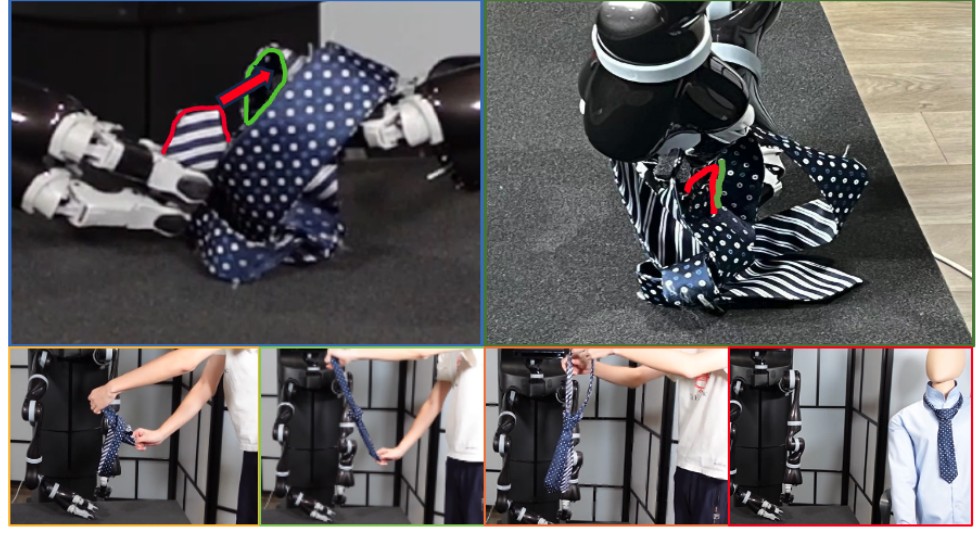

Figure C.9: Illustration of the success state of knotting a tie.

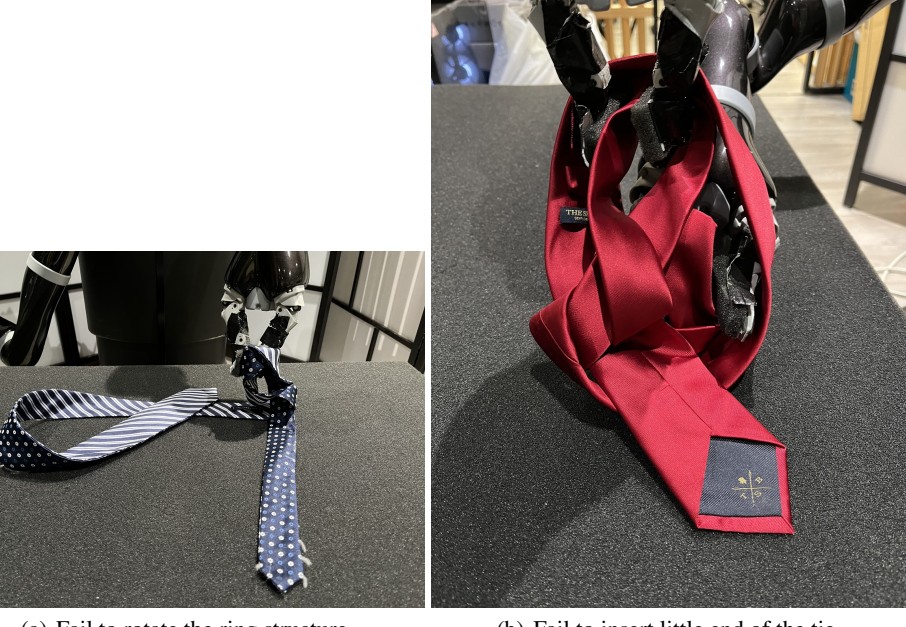

(a) Fail to rotate the ring structure.      (b) Fail to insert little end of the tie.

Figure C.10: Illustration of two major failure cases

## D    Additional Experiments on Towel-Folding Task

To illustrate our pipeline applies to other cloth manipulation tasks besides the tie-knotting task, we conduct experiments on the towel-folding task in the real setting.

### D.1    Real-World Experiment setup

We set up the real-world experiment with two robot arms as shown in Fig. D.11. The left arm is a Flexiv Rizon 4s arm and the right is a Flexiv Rizon 4 arm. One Realsense D435 RGB-D camera is mounted on the top of the aluminum tube. The size of the towel is 45cm × 75cm. This task requires two robot arms to fold the rectangular towel as shown in Fig D.12.

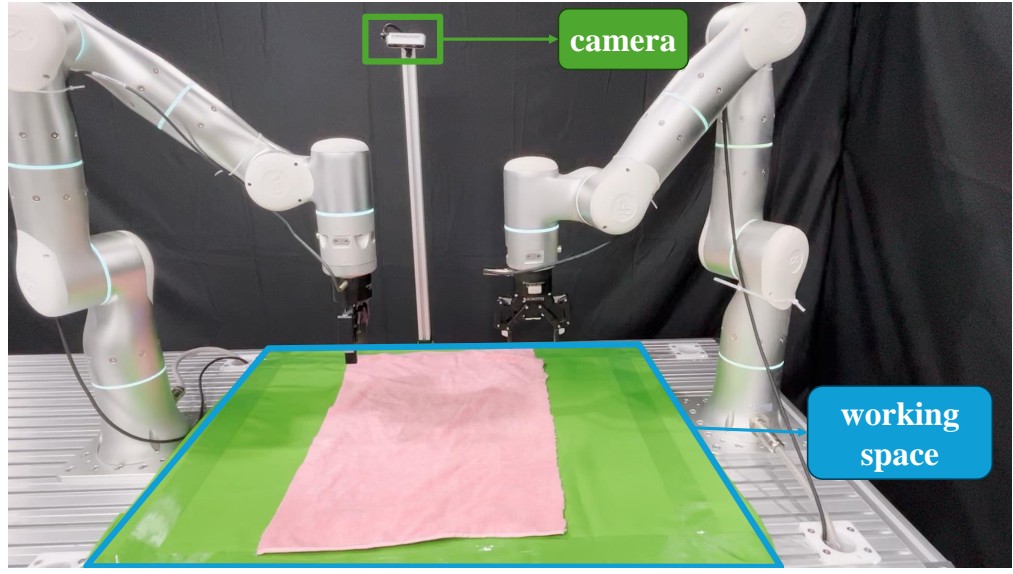

Figure D.11: Illustration of real-world experiment settings for towel-folding task.

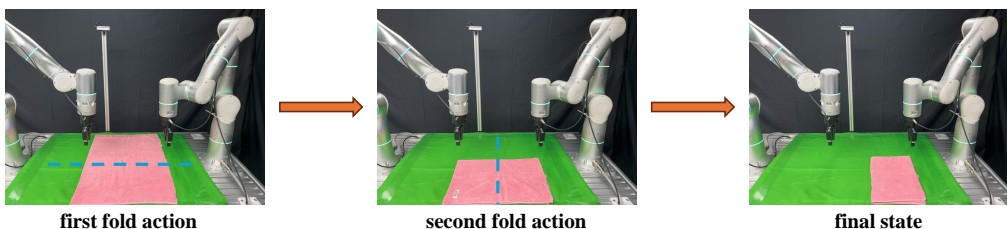

Figure D.12: Illustration of the process of the towel-folding task. The blue dash lines are the folding lines

