# OpenReview forum: "TieBot: Learning to Knot a Tie from Visual Demonstration through a Real-to-Sim-to-Real Approach"
_robot-learning.org/CoRL/2024/Conference — CoRL 2024_

### Official Review · Reviewer_eicD · 2024-06-27

**Originality:** 5
**Technical Quality:** 4
**Clarity Of Presentation:** 3
**Potential Impact:** 4
**Recommendation:** 4
**Confidence:** 4

**Review:**

Pro
1. Impressive tracking ability of the presented real2sim approach compared to video tracking under occlusions and hard-to-model cloth.
2. Extensive evaluation of the tracking approach (different initial poses of the tie, cloth material, and towel folding).
3. Real-world tie knotting experiment with a bi-manual robot.

Con
1. Lack of detail in some aspects of the pipeline (see Questions For Rebuttal).
2. Missing robot experiments for the towel folding experiment.
3. It is unclear how much the residual policy learning contributes and how it was implemented.

Minor comments:
1. Line 74-75: “To the best of our knowledge, this work is the first effort to develop a robot system, integrating perception, modeling, and robot learning” This claim feels very general and in its current form like an overstatement. For instance, here are three real2sim works that also perceive, model the real world, and learn a downstream policy:
- Reconciling reality through simulation: A real-to-sim-to-real approach for robust manipulation. Torne et al. RSS 2024
- URDFormer: A Pipeline for Constructing Articulated Simulation Environments from Real-World Images. Chen et al. RSS 2024
- ASID: Active Exploration for System Identification in Robotic Manipulation. Memmel et al. ICLR 2024

**Quality Of The Limitations Section:**

3

**Questions For Rebuttal:**

1. The reward function design in Equation 1 mentions variables C_1, C_2, C_3 but the experiment section never provides the concrete values being used. How much does the choice of those matter and how much tuning did you have to do?

2. The sim2real stage includes training a residual policy. Is this policy trained in the real world? Providing additional details, e.g., how many episodes were needed and how performance evolves over time would help better understand the sim2real gap.

3. The success rates in Table 1 show that knotting a tie from a single human demonstration video is still challenging. How do the failure cases look like especially w.r.t. to the different properties of the tie (unseen, softer, harder)?

4. Would collecting more than one demonstration benefit the state estimation and real2sim?

5. Line 296-297: “[The] pipeline still requires manually setting the initial state of the tie at the beginning of the Real2Sim stage.” How does this process look like and can it be automated?

**Robotics Focus:**

4

**Summary Of Paper:**

The paper proposes Hierarchical Feature Matching, combining correspondence-based local feature matching from RGB and learning to predict global keypoints from pointclouds. This novel state estimation method allows for real2sim transfer of the cloth’s state from a single human demonstration. The resulting sequence of meshes then act as subgoals for policy learning in simulation. To enable sim2real transfer, the authors utilize student-teacher distillation in simulation and residual policy learning in the real world.

**Summary Of Recommendation:**

The presented method shows impressive tracking ability in a real2sim setting as well as when executed on a real-world robotic platform. Minor improvements like real-world experiments on a different setup (e.g. towel folding instead of tie knotting) and adding ablations for the residual policy learning would further improve the paper.

---

### Official Review · Reviewer_XN7q · 2024-07-19
**Review of TieBot**

**Originality:** 3
**Technical Quality:** 4
**Clarity Of Presentation:** 3
**Potential Impact:** 3
**Recommendation:** 3
**Confidence:** 4

**Review:**

TieBot is a system for learning to tie knots via real to sim to real approach. The system involves first estimating subgoals of tying a knot from human video demonstrations by building correspondences with a mesh of the tie. These subgoals are then transferred to simulation and used to train a policy operating over privileged information, which is then distilled into a vision-based policy that is transferred to the real world.

The work is largely well written with a few clarification issues that I note in the question. The idea is simple and straightforward yet novel, using human video demonstrations to bootstrap simulation training and transfer back to the real world. The results are not particularly strong, the method only works half the time in the real world, but it is a fairly challenging task to operate from visual input so it is still impressive.

Strengths:
* evaluation on a complex manipulation task with first of its kind results
* thorough experimentation and analysis
* well written
* intriguing method for using human video to bootstrap simulation training, but also largely problem specific

Weaknesses:
* limited (though impressive) real world evaluation
* no real generalization, as far as I can tell it is trained and tested on the same tie
* method seems very specific to knot tying

**Quality Of The Limitations Section:**

3

**Questions For Rebuttal:**

1. Do you need to re-train a policy for each tie? Is the whole pipeline tie specific?
2. How much randomization is present in real world trials? I found "We test our real-world policy 10 times. Each time the initial positions of the tie are slightly perturbed about 5cm. The final success rate is 50%" but could not find a more detailed description in the paper or supplement.
3. What is the compute required to train the policies, how long does training take?
4. How are the grasp/pull actions defined, can the methods section be updated to make this much more clear?
5. 74-75: To the best of our knowledge, this work is the first effort to develop a robot system, integrating perception, modeling, and robot learning. This statement is clearly not true, perhaps the authors would like to amend it to a statement regarding knot tying?
6. Is the human arm segmented out in the intermediate steps of the video?
7. Can this method be applied to any other domains?

**Robotics Focus:**

4

**Summary Of Paper:**

This work proposes a system for learning to knot a tie by using human video to generate subgoals for supervising policy training in simulation which is then transferred onto real hardware.

**Summary Of Recommendation:**

I recommend accepting, I think this work has sufficient novelty and evaluation to meet the criteria for acceptance, though it would be helpful to clarify some of the minor concerns and comments.

---

### Official Review · Reviewer_NNyg · 2024-07-20
**Encouraging results in the challenging task of tie knotting from human demonstration video, but need more experimental results**

**Originality:** 3
**Technical Quality:** 3
**Clarity Of Presentation:** 2
**Potential Impact:** 3
**Recommendation:** 3
**Confidence:** 3

**Review:**

### Clarity:
- The clarity can be improved. I listed some below, but there might be more
- Table 2 has overlapping texts
- Citations should be added where necessary (e.g., CPD)
- There are places in the text spaces should be added (e.g., the title of 4.2, 3.1.3, mitigate Sim2Real gaps, places in appendix)
- Some consistency of text capitalization (e.g., title of 4.1)
- I suggest using a clearer notation in 3.1 so that readers can know which notations are the images, point cloud, meshes, and come from simulation or demonstrations. The current notation X refers to both point clouds and mesh sequences and is quite confusing
- The authors should talk about ATM first before using it in the Metrics section in 4.1
- In Figure 6, the authors should use a different color for the red cross sign to make it easier to see

### Originality:
- The paper has a certain level of novelty

### Significance:
- The paper tries to tackle a very challenging robotics problem with a proposed agent that can succeed half of the time in simulation

### Weaknesses:
- The results are encouraging for such a challenging task but still limited both in terms of the performance and the scope of the experiments (only one baseline is compared with and, two types of ties are used for experiments, limited ablation studies are provided)
- There are several manual steps in the proposed approach (e.g., manual key points definition), so it is unclear about the applicability of the method in other settings (there is some discussion in 4.3 about applying HFM to other cloth manipulation tasks but seems insufficient)

**Quality Of The Limitations Section:**

3

**Questions For Rebuttal:**

- The paper stated the student policy is trained using supervised learning; more details should be provided
- In Table 1, which result comes from simulation and which comes from real hardware?
- How do the authors connect PyBullet with DiffClothAI to train the student policy?
- In Table 2, does the result come from averaging all the meshes or just a single one, and what is the unit?
- It seems that it is pretty easy to experiment with the method for towel folding in the hardware; the authors are encouraged to add such an experiment instead of just showing the performance of HFM. This task can have more baselines to see the performance of the proposed method to state-of-the-art baselines

**Robotics Focus:**

4

**Summary Of Paper:**

The paper introduced a method for tie knotting from video demonstration containing of three phases. The first phase is to estimate mesh sequences from demonstration videos. The second phase use the estimated mesh sequences as subgoals to learn a policy in simulation to act from the point cloud. Finally, the third phase learn a residual policy to aid the simulation policy when deployed in the real.

**Summary Of Recommendation:**

The paper introduced an interesting method for learning tie knotting from human demonstration videos. The method has certain level of successes and advancement but more experiments should be performed and the paper's clarity should be improved.

---

### Author Rebuttal · Authors · 2024-08-14

We appreciate the time and efforts of all the reviewers and AC in reviewing our submission. We sincerely thank all the reviewers and AC for the constructive feedback and suggestions for further clarifying and improving our work.

Firstly, we are glad and thankful to know that the reviewers and AC are encouraged by the novelty, the thorough experimentation to tackle the challenging problem, and the encouraging and impressive results.  We really hope that our work can make reasonable and useful contributions and possibly may inspire future research on investigating how to use techniques.

In this comment, we focus on addressing the major/shared concerns and questions from the reviewers and the AC.

Q1. Application Scope and Real-World Evaluation (Reviewer NNyg, XN7q, eicD) (see official comment below)

Q2. Amend the Claim to be More Specific (Reviewer XN7q, eicD)  (see official comment below)

We also addressed other comments/questions/issues raised up by each specific reviewer by directly replying to each reviewer.
We have revised the main paper, the supplementary document, and the video, to incorporate major changes. We would be very happy to incorporate any other requested changes in the final paper if accepted.

Thanks again for your valuable reviews/comments!

Best regards,

Authors

---

### Decision · Program_Chairs · 2024-09-04

**Decision:**

Accept

**Comment:**

The paper introduced a method for tie knotting from video demonstration containing three phases. The first phase is to estimate mesh sequences from demonstration videos. The second phase uses the estimated mesh sequences as subgoals to learn a policy in simulation to act from the point cloud. Finally, the third phase learns a residual policy to aid the simulation policy when deployed in the real.
The paper deals with the very challenging manipulation tie-knotting problem, is well-written, and provides thorough experimentation and analysis especially its tracking ability.

The reviewers had originally important comments and concerns on the specificity of the approach and its components to the tie-knotting problem, the limitations in terms of its performance, and the limited scope of the experiments on generalization for the task and for the object (tie).

While the authors well-addressed the comments of the reviewers, I agree that more experimental evaluation with the real robot should be included in a robotics paper. Therefore I recommend the paper to be accepted for poster presentation.